# Attenuation of SCI-Induced Hypersensitivity by Intensive Locomotor Training and Recombinant GABAergic Cells

**DOI:** 10.3390/bioengineering10010084

**Published:** 2023-01-09

**Authors:** Stanislava Jergova, Elizabeth A. Dugan, Jacqueline Sagen

**Affiliations:** Miami Project to Cure Paralysis, Miller School of Medicine University of Miami, 1095 NW 14th Terrace, Miami, FL 33136, USA

**Keywords:** spinal cord injury, chronic pain, intensive locomotor training, recombinant cells, GABA, NMDA, serine-histogranin

## Abstract

The underlying mechanisms of spinal cord injury (SCI)-induced chronic pain involve dysfunctional GABAergic signaling and enhanced NMDA signaling. Our previous studies showed that SCI hypersensitivity in rats can be attenuated by recombinant rat GABAergic cells releasing NMDA blocker serine-histogranin (SHG) and by intensive locomotor training (ILT). The current study combines these approaches and evaluates their analgesic effects on a model of SCI pain in rats. Cells were grafted into the spinal cord at 4 weeks post-SCI to target the chronic pain, and ILT was initiated 5 weeks post-SCI. The hypersensitivity was evaluated weekly, which was followed by histological and biochemical assays. Prolonged effects of the treatment were evaluated in subgroups of animals after we discontinued ILT. The results show attenuation of tactile, heat and cold hypersensitivity in all of the treated animals and reduced levels of proinflammatory cytokines IL1β and TNFα in the spinal tissue and CSF. Animals with recombinant grafts and ILT showed the preservation of analgesic effects even during sedentary periods when the ILT was discontinued. Retraining helped to re-establish the effect of long-term training in all of the groups, with the greatest impact being in animals with recombinant grafts. These findings suggest that intermittent training in combination with cell therapy might be an efficient approach to manage chronic pain in SCI patients.

## 1. Introduction

Spinal cord injury (SCI) and the associated pathologies present a very complex clinical problem. The loss of motor functions is the most serious and debilitating outcome, however, secondary pathologies contribute significantly to a low quality of life in SCI patients. Chronic pain following an SCI is clinically challenging, with there being insufficient treatment options [1,2,3,4,5,6,7]. The underlying pathologic mechanisms affect several levels of pain transmission from the periphery to the central locations, both in the ascending and descending pathways. Most of the currently available analgesics are designed to target single specific signaling sites, omitting other potentially involved mechanisms. There is growing data on the management of chronic pain suggesting that multitargeted approaches are the most efficient treatments [8]. 

One of the key underlying mechanisms of chronic pain development is disinhibition, which is caused by dysfunction of γ-aminobutyric acid (GABA) signaling and disruption of inhibitory balance in the spinal pain processing sites [9,10,11,12,13,14,15,16,17,18]. Previous studies demonstrated that disinhibition can be modified by an intraspinal injection of neuronal GABAergic precursors [19,20,21,22,23,24,25,26,27,28,29,30,31]. This approach showed reduced pain-related behavior in models of peripheral nerve injury and SCIs. In order to enhance the analgesic effects of the cell-based treatment, we have generated recombinant GABAergic cells expressing N-methyl-D-aspartate (NMDA) antagonist serine-histogranin (SHG). The hyperexcitability of spinal neurons due to enhanced glutamate release from peripheral afferents and the activation of dorsal horn NMDA receptors is considered to be another key event in the process of chronic pain development and maintenance. NMDA antagonists have been shown to produce analgesic effects on several experimental pain models [32,33,34]. However, the widespread distribution of GABA and NMDA receptors in the central nervous system (CNS) limits their use via systemic treatments. To avoid systemic side effects, local targeted approaches are needed. Thus, recombinant cell-based therapy targeting both GABA and glutamate signaling might be more beneficial than individual strategies are. In fact, our studies demonstrated the beneficial effects of such treatment in models of peripheral nerve injury and SCI-induced pain. The intraspinal injection of engineered cells improved and prolonged the analgesic effects of cell-based treatment of chronic neuropathic pain [20,30,35]. 

Physical exercise is an important approach in the overall rehabilitation strategies post-SCI, and numerous clinical and preclinical studies have pointed to its beneficial effect in attenuating pain in animal models and SCI subjects [31,36,37,38,39,40,41,42,43]. One of the hypothesized mechanisms is the capability of exercise intervention to reduce the chronic inflammation induced by SCI [44,45,46,47,48]. Our previous studies showed that intensive locomotor training (ILT) in animals with an SCI reduced or prevented the development of pain-related behavior, and that the combination of locomotor training and cell-based therapy with GABAergic progenitors significantly enhanced the analgesic outcome of such an approach [49]. The current study is built upon these findings, and evaluates the potential for recombinant GABAergic cells in combination with exercise to maximize the beneficial effects of both treatments in reducing chronic SCI pain. In addition, the capability of our treatment to preserve analgesic effects during off-training periods was also evaluated. The preliminary findings were presented at the American Pain Society meeting and the Society for Neuroscience meeting [50,51].

## 2. Materials and Methods

*Animals:* Adult male Sprague Dawley rats were used in all of the experiments (140–160 g at the time of the first surgery); pregnant female Sprague Dawley rats were used for E14 embryo harvesting (Envigo, Indianapolis, IN, USA). Two animals were housed per cage with free access to food and water in a 12 h light/dark cycle. The experimental procedures were reviewed and approved by the University of Miami Animal Care and Use Committee, and we followed the recommendations of the “Guide for the Care and Use of Laboratory Animals” (National Research Council). All of the surgical procedures were conducted under 2.5% isoflurane/O_2_ anesthesia using aseptic conditions. All of the animals were randomly assigned to the experimental groups in accordance with the ARRIVE guidelines [52]. The order of the animals undergoing the surgeries and behavioral evaluations was also randomized. All of the behavioral tests were performed by experienced investigators who were blinded to the treatments. A power analysis was used to calculate number of animals per group using SigmaPlot 15.0.0.13. software (Inpixon, Palo Alto, CA, USA). The data from our previous studies using the same SCI model and training regimen were used to calculate the sample size. Values used were as follows: minimum detectable difference in the means = 1.3, the expected standard deviation of the residuals = 0.6, number of groups = 6; based on these data the suggested sample size was 7. This number was further increased to include additional animals for the histological and biochemical analyses. For the retraining experiment, input values used were as follows: minimum detectable difference in the means = 1.2, expected standard deviation of the residuals = 0.4, number of groups = 3; the suggested sample size was 4. Only the animals that were able to fully recover from the initial surgery without prolonged health issues and that displayed signs of hypersensitivity post-SCI were included in the study. The total N = 84; the N’s for the individual experiments and treatment groups appear in the figure legends. A schematic representation and timeline of the overall study can be found in Appendix A.

*Induction of spinal injury:* The spinal cord clip injury model [53] has been successfully used in our lab to establish SCI-induced hypersensitivity [30,35,54,55,56,57,58,59]. Briefly, under isoflurane anesthesia, the back of the rat was shaved and cleaned using alcohol pads. A laminectomy was performed to expose the spinal cord segments T6–T8. An aneurism clip 1 mm wide (20 g compression force; Harvard Apparatus, Holliston, MA, USA) was oriented in a vertical position and the spinal segment in the area between T6–T7, and it was compressed for 60 s without disturbing the dura or dorsal roots. After the procedure, the incision was closed, and the rats were returned to their home cages. Sham animals underwent the same procedure, but we omitted the clip injury. The bladder was manually expressed in the SCI animals for 7–10 days, or until voiding was regained. The rats were tested for up to 15 weeks after the spinal compression.

*E14 cell isolation and generation of recombinant cells:* E14 Sprague Dawley rat embryos were dissected from pregnant dams, placed in ice-cold Hank’s Balanced Salt Solution (HBSS), and the lateral ganglionic eminences were removed. The tissues were pooled, centrifuged at 4 °C, resuspended in HBSS, dissociated by trituration, and single-cell suspensions were collected. The dissociated neural precursor cells (NPCs) were pelleted and re-suspended in Dulbecco’s Modified Eagle Medium with nutrient mixture F12 (DMEM/F12) and with 1% N2 supplement and 10 ng of fibroblast growth factor 2 (FGF-2)/mL; R&D Systems, Minneapolis, MN, USA). The NPC cultures were seeded in tissue culture flasks at 1 × 10^5^ cells/cm^2^. NPCs formed neurospheres over 3 or 4 days in vitro. The withdrawal of FGF-2 on 1 day prior to transplantation was used to induce the pre-differentiation of NPCs to the GABAergic phenotype [19,60]. To generate recombinant cells, adeno-associated virus (AAV) serotype 2/8 encoding green fluorescent peptide (GFP) or SHG cDNA were prepared by the Miami Project Viral Vector Core. SHG cDNA was previously engineered in our lab (see [30] for details). The cells were transduced by AAV vectors 24 h after harvesting. Briefly, the cells were pelleted at 1000 rpm for 5 min, and 10 µL of purified virus (1 × 10^9^ transfection units/mL) was added to a minimal volume of culture media for 4–5 h at 37 °C. After washing the pellet to remove the unbound virus, the cells were plated with an appropriate amount of culture media. The production and secretion of recombinant peptide were verified by immunocytochemistry and a fluorescence-linked immunosorbent assay (FLISA) analysis.

*Immunocytochemistry and FLISA analysis of cell cultures:* For the phenotypic characterization of the naïve and recombinant NPCs, standard immunocytochemical protocols were used. The cells were plated in poly-L-ornithine/fibronectin coated Lab-Tek (Nunc, ThermoFisher, Waltham, MA, USA) chambers at a concentration of 5 × 10^5^/well and incubated at 37 °C for 2–3 days, fixed with 4% paraformaldehyde, washed and incubated in 5% normal goat serum for 2 h. Primary antibodies (anti-GABA (1:200), anti-Glutamate (1:1000, both Millipore, Burlington, MA, USA), anti-SHG (1:50, custom synthesized by 21st Century Biochemicals, Marlborough, MA, USA) were added for 24–72 h of incubation, which was followed by washing and incubating them with appropriate secondary antibodies (anti-mouse and anti-rabbit and anti-guinea pig AlexaFluor 488 and 597, 1:250, Invitrogen, Burlington, MA, USA) and DAPI for nuclear staining. After a final washing step, the upper structure of the chamber was carefully removed, and the slides were air-dried and coverslipped (Vectashield, VectorLabs, Newark, CA, USA). To estimate the transduction rate, at least 4 images of areas with clearly identified single cells per well labeled with SHG and DAPI were captured using a confocal microscope (Spectral Confocal Microscope Fluoview1000, Olympus, Center Valley, PA) and analyzed using ImageTool software by manual counting based on overlapping single-colored images and the colocalization of the markers. The DAPI-labeled objects with a diameter that was under 8 µm were not included in the counting as they may have been artifacts of staining or apoptotic nuclei.

The FLISA method was used to quantify the amount of SHG in the cell supernatants, as previously published [30]. Briefly, 100 µL of samples with the same amount of peptide were loaded into 96-well plates in capture buffer according to the manufacturer’s protocol (Odyssey FLISA, Li-Cor, Lincoln, NE, USA) and incubated overnight at 4 °C, which was followed by incubation in the blocking buffer for 3 h at room temperature and an anti-SHG primary antibody (1:100, 21st Century Biochemicals) for 18 h at 4 °C. The plates were then washed and incubated in a secondary antibody (IRDye goat anti-rabbit) for 2 h, washed, and absorbance measured using an Odyssey Infrared Imager (Li-Cor, Lincoln, NE, USA). The SHG peptide at different concentrations was used to calculate the standard curve and the relative amount of SHG in the samples. The data were analyzed by Graph Prism polygnomial analysis.

*Transplantation*: The cells were injected into the lumbar enlargement to target below-level neuropathic pain, as is frequently reported by SCI patients [1,61,62,63,64]. Changes in the nociceptive signaling in remote areas of the spinal cord in relation to the injury site has also been reported in several studies, and this may underlie the development of below-level pain [65,66,67,68]. At 4 weeks post-SCI, when hypersensitivity was clearly established, the animals were intraspinally injected with non-recombinant or recombinant cells. For this, the rats were anesthetized, and a laminectomy was performed aseptically at the Th13-L1 vertebrae to expose the lumbar enlargement. Two μL of cells (100,000 cells/µL) were loaded into a glass micropipette attached to a Hamilton syringe and injected bilaterally using a stereotaxic framing stage (Stoelting, Wood Dale, IL, USA) at a depth of 0.5 mm from the dorsal surface and at a rate of 0.5 μL/min. The needle was left inserted in the tissue for another 1–2 min post-injection to avoid backflow. The muscles were then sutured in layers, the overlying skin was closed using staples, and the animals were transferred to heated cages for recovery. All of the transplanted rats received cyclosporine A (IP, 10 mg/kg; Bedford Labs, Bedford, MA) from day 3 until 2 weeks post-SCI, and then, they remained on oral administration (45 mg/kg) for the duration of the study.

*Exercise protocol:* Intensive locomotor training (ILT) on a treadmill (Columbus Instruments, Columbus, OH, USA) was initiated at 5 weeks post-SCI to target the chronic hypersensitivity that developed post-injury. The animals underwent 5 days/week of training with a ramping-up protocol [49]. Briefly, the protocol starts with a 10m/min pace for 5 min, which increased in speed (+1 m/wk) and time (1–4 min/wk) to a maximum of two 20 min sessions/day at a 15m/min pace by the 4th week depending on the individual animal’s locomotor abilities. A minimum of a 10 min break was given between 2 consecutive daily sessions. An 8 degree incline on the treadmill was used, requiring a higher load to be placed on the hind limbs during the exercise, which increased the intensity and their heart rate and is beneficial for locomotor improvement following an SCI [69,70,71,72]. *Retraining:* To evaluate the sustainability of the effects of the exercise with/without grafts on the behavioral and biochemical outcomes of the treatments, subgroups of animals underwent intermittent sedentary and retraining periods. After 10 weeks of the standard ILT protocol above, SCI animals from the trained group without the graft (ILT), trained animals with the naïve NPC grafts (NPC+ILT), and trained animals with recombinant grafts (rNPC+ILT) were kept for an additional 5 weeks in a sedentary (no ILT) condition, and then retrained for 5 weeks following the same protocol as above. For comparison, a group of SCI animals from the ILT, NPC+ILT, and rNPC+ILT groups that were also left in the sedentary condition for 5 weeks did not receive the retraining (thus, they remained sedentary for 10 weeks after the initial ILT training). A second subgroup of animals from the retrained cohort was further observed and retrained after an additional 3 week sedentary period, which was followed by an additional 2 weeks of retraining.

*Behavioral analysis:* The animals were evaluated for the presence of SCI-induced hypersensitivity to tactile, heat, and cold stimuli [4,7] weekly for up to 15 weeks post-SCI, and then once at the end of each sedentary and retraining period. All of the behavioral testing was conducted in a designated animal testing room, and was performed during the same time of day. *Tactile hypersensitivity:* Calibrated von Frey hairs ranging from 0.4 to 15 g were used to determine tactile hypersensitivity [73]. The rats were placed in an elevated plastic cage with wire mesh floor and allowed to acclimate for about 15 min. Von Frey filaments were gently pressed on the plantar hind paw for about 6 s. A brisk withdrawal from the filament in conjunction with a supraspinal response (e.g., turning head towards the stimulus, walking away from the stimulus, vocalization, licking, and shaking the paw) was counted as positive response, and the next lower filament was tested. A total of six responses were recorded, and these were used to calculate the withdrawal threshold [74]. Generally, uninjured rats do not respond to a force of less than 15 g. An upper limit of 15 g was used in most experiments since a greater force may lift the hind paw itself.

*Cold hypersensitivity:* Sensitivity to a non-noxious cooling stimulus was evaluated using acetone. One hundred µL of acetone was dropped onto the plantar surface of the hind paw from a blunted 22 g needle attached to a syringe, and the response was recorded. A positive response consisted of a brief paw withdrawal accompanied by at least one supraspinally mediated reaction, such as licking, shaking the paw, or moving away from the stimulus. Acetone was applied to the hind paw 5 times, with there being 1–2 min between the applications. The total number of positive responses was converted to a percent response frequency [75]. In normal, intact rats, acetone does not evoke a withdrawal response.

*Heat hypersensitivity:* The Hargreaves test was used to assess heat hypersensitivity [76]. The rats were placed in a clear plastic cage on an elevated glass floor with a radiant heat source (52 °C) beneath the glass that was aimed at the plantar hind paw (Ugo Basile, Chicago, IL, USA). A timer was activated together with the infrared beam and stopped upon animal’s paw withdrawal. Withdrawal latencies that were clearly elicited by the increasing heat (paw withdrawal with another supraspinal responses as above) and not by the random movement of animal in the cage were recorded in 3 trials for each hind paw with there being at least 30 s of rest between the trials. *BBB for functional recovery assessment*: The level of locomotor recovery was evaluated weekly throughout the study using the BBB locomotor rating scale [77]. The animals were placed into the open field and observed by an experienced investigator for about 4 min. The position of the paws, trunk, tail, and coordination were scored using a 21 point BBB scoring system, where 0 indicates complete paralysis and a score of 21 indicates normal locomotion.

*Intrathecal catheter implantation and injections:* To evaluate the contribution of GABAergic or recombinant peptide from the cell grafts and ILT to the behavioral outcomes, GABA-A receptor antagonist bicuculline or SHG antibody or both were injected via intrathecal catheter following 10 weeks of the ILT, and the animals were re-tested for the reversal of the antinociceptive effects. Indwelling catheters (7.5–8 cm; ReCathCo, Allison Park, PA, USA) were inserted through a slit in the atlanto-occipital membrane down the intrathecal space and secured to the neck muscles with sutures, similar to previously described methods [78]. The rats were allowed to recover for at least three days following the intrathecal surgery prior to their use in the experiments. The drug administrations were randomized with at least a 48 hr washout period between different drugs. The SHG antibody (0.5 µg/mL, 21st Century Biochemicals) or bicuculline (0.3 µg; Sigma/Millipore, Burlington, MA, USA) were dissolved in saline and injected in 5 µL volumes, which was followed by a 5 µL flush with saline. For the mixed injections (bicuculline and SHG antibody), 5 µL of each drug was injected [30,59]. Since the behavioral outcomes after the 10 weeks of ILT differ between the treatment groups, we have normalized the individual data for each animal to the pre-injection values (after 10 weeks of ILT), and we express the results of drugs as fold changes from their pre-injection values.

*Tissue processing:* At the termination of the behavioral studies, the animals were deeply anesthetized by CO_2_ inhalation and either transcardially perfused with 4% paraformaldehyde followed by saline for immunohistochemistry or dissected to harvest the fresh spinal tissue and cerebrospinal fluid (CSF) for the biochemical analysis. The spinal cord tissue was dissected, post-fixed overnight in 4% paraformaldehyde, and cryoprotected in 30% sucrose. The tissue was sectioned on cryostat and either free-floating or slide-mounted sections were collected in a serial manner. Fresh spinal cord tissue used for the biochemical analysis was dissected and frozen on dry ice, homogenized, and stored at −80 °C.

*Biochemical analysis:* Enzyme-linked immunosorbent assay (ELISA) kits were used to analyze the presence of various cytokines in the CSF and lumbar spinal dorsal horn tissue. The tissue was homogenized, and the protein concentration in the homogenates and CSF samples was established by the BCA method (Thermo Fisher, Waltham, MA, USA). Samples at the same protein concentration were then used for ELISAs following the manufacturer protocols. Briefly, samples were loaded into pre-coated 96-well plates provided by the kit, incubated with the respective primary and secondary antibodies, washed, and the optical density was measured on a microplate reader (Molecular Devices, San Jose, CA, USA) at the wavelength that was suggested by the kit. The final concentrations of cytokines in the samples were calculated based on the standard curve run with each experiment (SoftMax software, Molecular Devices, San Jose, CA, USA). FLISA: The presence of SHG in the spinal tissue and CSF was analyzed by FLISA. The samples of tissue or CSF with the same protein content were loaded into 96-well plates, and the same protocol as that used for cell analysis (above) was followed.

*Immunostaining:* Standard immunostaining protocols were used to visualize the dorsal horn following the various treatment interventions. The spinal cord tissue sections were incubated in 5% normal goat serum (NGS) in 0.1M phosphate-buffered solution with 0.05% Triton X detergent (PBS-Tx) for 2 h, which was followed by overnight incubation with the primary antibody (neuronal nucleus (NeuN), 1:1000; GABA, 1:200; glial fibrillary acidic protein (GFAP), 1:1000 (all Sigma/Millipore, Burlington, MA, USA); SHG, 1:50 (20th Century Biochemicals); ionized calcium-binding adapter molecule 1 (Iba-1), 1:500 (Wako/Fujifilm, Madison, WI, USA); Doublecortin, 1:200 (Abcam, Waltham, MA, USA) in 5% NGS in PBX-Tx. The following day, the tissue was washed with PBS and incubated with appropriate secondary antibodies (Alexa Fluor anti-rabbit, anti-mouse, 1:250, Invitrogen/Millipore). We used 4′,6-diamidino-2-phenylindole (DAPI, 1:1000, Sigma/Millipore, Burlington, MA, USA) for the nuclear staining. The slides were air-dried, cover-slipped, and evaluated using the fluorescence microscope and image analysis software (Image J, NIH).

*Quantification of grafted cells in the spinal tissue:* The grafted cells were identified as being GFP or SHG positive. Colocalization with NeuN, GABA, Iba-1, GFAP, and Doublecortin was evaluated under 20× magnification in at least 10 serial sections from the lumbar region per animal (3 animals per group), with a 160 µm gap between the sections. The counts were performed bilaterally, and the average number of grafted cells and the percentage of co-expression of two markers were compared between the groups.

*Image analysis:* For quantification, the analyses were performed using a Zeiss Axiovert 200M research microscope furnished (Ludl Electronic Products, Hawthorne, NY, USA) with a DVC cooled camera and multi-band fluorescent filters, allowing for viewing single, double, or triple fluorophores at the same time. The best examples of double labeling were analyzed using the confocal microscope (Olympus Spectral Confocal Microscope Fluoview 1000, Olympus, Center Valley, PA, USA).

*Statistical analyses:* The data are expressed as mean ± S.E.M. The behavioral data over time were analyzed by two-way ANOVA between the groups and time post-injury, with time being taken as a repeated measure, with a Tukey post hoc analysis. The single time point behavioral data (at the 15 weeks post-SCI or after the retraining/sedentary periods) and biochemical data were analyzed by ANOVA, with individual comparisons being made by post hoc tests utilizing the Newman–Keuls correction where appropriate. The behavioral data from the pharmacological experiments were normalized, and they are expressed as the fold change from the 10 weeks ILT data. A t-test was used to compare the cell counts for the immunocytochemical studies. The statistical analyses were performed using SigmaStat (Inpixon, Palo Alto, CA, USA). Statistical significance was taken at *p* < 0.05.

## 3. Results

### 3.1. Characterization of Recombinant GABAergic NPCs

The phenotype of the NPCs and the presence of recombinant peptide SHG were evaluated by immunocytochemistry and FLISA. Numerous cells developed into a GABA-expressing phenotype. SHG was detected in about 70% of the cells after transduction with AAV2/8-SHG (Figure 1a). The presence of SHG in the cell supernatant was evaluated by FLISA, with there being no detectable SHG in the non-recombinant cell cultures as expected (*p* < 0.001 rNPC vs. NPC, Figure 1b). A few glutamate-labeled NPCs were detected in the cultures (Appendix A).

### 3.2. Behavior

Summary observations: The ILT alone improved the overall pain-related behavior in each of the behavioral tests. Cell grafting also improved the pain-related behavior, with there being better outcomes in the animals with the recombinant grafts. The combination of training and cell grafts appeared to be the most potent approach, with it having stronger effects on the animals with recombinant grafts. Further differences between the groups were observed when the training was discontinued for several weeks and the animals were retrained for a short time. Retraining helped to boost the analgesic effects of the previous ILT and, in combination with recombinant cells, it helped to prolong the analgesic effects on the treated animals.

#### 3.2.1. Locomotor Score

The BBB score across the groups were not affected by any of the treatments. The animals showed partial recovery of the hind limb paralysis starting from 1 week post-injury. The locomotor scores plateaued at around 12 points (Appendix A).

#### 3.2.2. Tactile Hypersensitivity

SCI resulted in the gradual development of tactile hypersensitivity that persisted throughout the experiment in the SCI-only animals. The withdrawal threshold was reduced by 2 weeks post-injury in all of the groups compared to the baseline, with the *p* being at least <0.05 (Figure 2a; for better clarity of the plot, we only labeled this time point since all of the animals reached a significantly reduced threshold compared to their baseline values). The intraspinal injection of recombinant and nonrecombinant cells, the ILT, and their combination improved the behavioral outcomes of all of the treated animals compared to those with SCI only, with the *p* being at least <0.01, starting from 2 weeks post-grafting/1 week post-ILT initiation and lasting till the end of the experiment (Figure 2a, overall F(df5,736) = 47.11, *p* < 0.0001). The individual groups showed differences at earlier time points as well; for better clarity of the plots, we omitted additional marks for the time-course plots, and rather, we used the end-point values to compare the data between the treatment groups. The final values (15 weeks post-SCI, Figure 2b) comparison showed differences from those of the SCI-only group with all of the treated animals (*p* < 0.01 and 0.001). Additionally, the animals with recombinant grafts showed better outcomes compared to those of their nonrecombinant graft cohorts (*p* < 0.05). The combination of the recombinant graft with training was the most effective treatment overall, with a significantly better outcome compared to those of both the ILT alone and NPC+ILT groups (*p* < 0.05). This rNPC+ILT group reached a nearly complete reversal of the tactile hypersensitivity compared to the pre-injury baseline responses towards the endpoint of the study.

#### 3.2.3. Heat Hypersensitivity

SCI induced the development of heat hypersensitivity with a progressive drop in the withdrawal latency in all of the SCI animals (Figure 2c), with the *p* being at least <0.01 compared to pre-injury baseline starting at 2 weeks post-injury in all of the groups (for better clarity of the plot, we only labeled this time point since all of the animals reach significantly reduced threshold compared to their baseline values). Treatment with the cell grafts, the ILT, and their combination improved the outcome in all of the treated animals compared to that of SCI only, with the *p* being at least <0.01, as observed at 5 weeks post-grafting/4 weeks post-ILT initiation (individual groups showed differences at the earlier time points as well; for better clarity of plots, we omitted additional marks for the time-course plots, and rather, we used the end-point values to compare the data between the treatment groups), and lasting until the end of the experiment (Overall F(dF5, 864) = 44.01, *p* < 0.0001). The endpoint values (Figure 2d) showed improvement in all of the treated groups compared to those of the SCI-only animals (*p* < 0.01 and 0.001). The animals treated with rNPC+ILT appeared show the most robust effects, having a significantly better outcome compared to the ILT-only and rNPC-only groups (*p* < 0.05). Again, this rNPC+ILT group reached a near complete reversal of the heat hypersensitivity compared to the pre-injury baseline responses towards the endpoint of the study.

#### 3.2.4. Cold Hypersensitivity

The SCI induced cold hypersensitivity by 2 weeks post-injury in all of the animals (Figure 2e). In the SCI-only group, the hypersensitivity gradually increased, and this was sustained at around 70% until the end of the experiment. The cell grafts, ILT, and their combination significantly reduced the cold hypersensitivity, with the *p* being at least <0.01 for all of the treated animals, as observed from 2 weeks post-grafting/1week post-ILT initiation (overall F(dF5, 864) = 102.80, *p* < 0.001). The endpoint values showed significant reduction of the cold hypersensitivity in all of the treated groups compared to those who underwent the SCI only (*p* < 0.001, Figure 2f), with there being no significant differences between the treatment interventions, which was likely due to the basement effect.

#### 3.2.5. First Retraining

Subgroups of animals were further evaluated after 5 weeks of a sedentary period, which was followed by either an additional 5 weeks of retraining or 5 weeks with no retraining. The behavioral testing after the first sedentary period in all of the groups showed a partial reduction of the analgesic benefits of the treatments compared to outcomes at the end of the original 10 week course of ILT. Retraining helped them to regain some of the analgesic benefits, especially in the groups with recombinant grafts. In the “no retraining” cohort, generally, the reduction of the analgesic benefits further continued.

*Tactile hypersensitivity:* In the retraining cohorts (Figure 3a), the reduction of the paw withdrawal threshold (PWT) after a sedentary period was observed in all of the groups compared to the values after 10 weeks of the ILT (*p* < 0.05 for the ILT and rNPC+ILT; *p* < 0.01 for the NPC+ILT). Although it was not significant, there appeared to be less decline in the PWT during the sedentary period in the animals with the rNPC grafts than in the control NPC grafts. Retraining improved the PWT in all of the groups, partially restoring the initial beneficial effects of the ILT, with there being a significantly better outcome in the rNPC+ILT group compared to that in the ILT-only group (*p* < 0.05). In particular, in the rNPC+ILT group, the PWT following retraining was nearly restored to the levels that were obtained following the initial ILT regimen. In the no retraining cohorts (Figure 3b), there was a progressive reduction of the PWT in all of the animals after this prolonged sedentary period compared to that following the initial 10 weeks of ILT training (*p* < 0.05 for all groups). The animals grafted with rNPC appeared to show a more prolonged retention of the ILT effects.

*Heat hypersensitivity*: In the retraining cohorts (Figure 3c), the withdrawal latency of the animals during the first sedentary period decreased in all of the groups. However, there was a significant difference in the latency values during this period between the groups. In the ILT-only group, the initial values were lower compared to other groups, and the further reduction of the latency observed during the sedentary period was not significant compared to that group’s low values following the initial ILT regimen. In the NPC+ILT group, the initial post-ILT values were higher, and the drop during the sedentary period was significant (*p* < 0.05). In contrast, in the rNPC+ILT group, while the initial post-ILT values were also high, there was greater retention of the increased latency scores, and the drop during the sedentary period was not statisically significant. The latency values at this point were also different (higher) in the rNPC groups compared to the latencies in the other groups, suggesting a possible positive effect of rNPC in preserving the antihyperalgesic effects even during the sedentary period (*p* < 0.05 vs. both ILT only and NPC+ILT). Retraining after the sedentary period improved the overall outcome in all of the groups, almost to levels comparable to their corresponding initial values. There was significant improvement observed in the NPC+ILT group with *p* < 0.05 compared to the sedentary period within this group. Following retraining, only the rNPC+ILT group showed a significantly increased latency compared with animals that underwent the ILT alone (*p* < 0.05). In the non-retrained cohorts (Figure 3d), the withdrawal latencies progressively decreased over time (*p* < 0.05 for all of the groups), although the latency after the prolonged sedentary period in the grafted groups was still higher compared to those of the non-grafted group (*p* < 0.05 for NPC+ILT and rNPC+ILT compared to ILT only).

*Cold hypersensitivity:* The beneficial effects of the ILT and cell grafting were most substantially observed for cold hypersensitivity, reaching near maximal effects following 10 weeks of the ILT for all of the treatment groups. In the retraining cohorts (Figure 3e), after the first sedentary break, a significant increase in cold hypersensitivity was observed in the animals who underwent the ILT alone and in the NPC+ILT groups (*p* < 0.01 compared to the initial 10 weeks of ILT). The values in the rNPC+ILT group also modestly increased during this sedentary period, but not significantly, again suggesting better preservation of the antinociceptive effects of the ILT with the recombinant grafts. Following the sedentary period, both of the graft groups showed significantly lower cold hypersensitivity than the non-grafted ILT-only group (*p* < 0.05 for NPC+ILT and *p* < 0.01 for rNPC+ILT compared to ILT only). After the retraining, the cold scores dropped down to their initial levels in all of the groups. In the non-retrained cohorts (Figure 3f), the grafted groups showed less reversion to the SCI cold hypersensitivity after the prolonged sedentary period. In the ILT-only group, there was a sustained increase in the hypersensitivity throughout the 10 weeks sedentary period, with *p* < 0.05, compared to the initial ILT period. In the NPC+ILT and rNPC+ILT, an increase in the cold hypersensitivity was observed over the 10 weeks sedentary period, with *p* < 0.05, compared to their initial values, however, the values were significantly lower compared to those the ILT-only group (*p* < 0.05).

#### 3.2.6. Second Retraining

Due to compelling findings in the retraining study, a small additional study group (n = 4/group) from the retraining cohorts were further evaluated for the presence of hypersensitivity after another 3 week sedentary period, which was followed by 2 weeks of a second retraining. Despite the low numbers, there was a similar trend in strengthening the analgesic benefits after a short period of second retraining, with the best outcomes being observed in the animals with recombinant grafts.

*Tactile hypersensitivity* (Figure 4a): A second sedentary break led to another drop in the PWT, and although it was not significantly different from the values following the first sedentary period (perhaps due to the small number of animals), there appeared to be a tendency towards a higher PWT compared with that of the first sedentary period. The second retraining also led to increases in the PWT in each treatment group, which were comparable with their first retraining data. Overall, the values in the rNPC group fluctuated less between the repeated ILT and sedentary periods throughout the time course (*p* > 0.05 within the various periods in the rNPC group). This rNPC group also showed a significantly better outcome after the second retraining compared to the ILT-only group (*p* < 0.05), suggesting the possible improved preservation of the established antinociceptive effects in the group with recombinant grafts.

*Heat hypersensitivity* (Figure 4b): The second retraining showed similar patterns across the groups, with an attenuation of the antinociceptive effects of the ILT during the first and the second sedentary periods (*p* < 0.05) and the reinstatement of these beneficial effects after both of the retraining sessions. A comparison of sedentary values across the groups showed that the rNPC group preserved significantly higher latencies compared to the other two groups during both of the break periods (*p* < 0.05 for the first sedentary period compared to the ILT-only group and for the second sedentary period compared to both ILT only and NPC+ILT groups).

*Cold hypersensitivity* (Figure 4c): The beneficial effects of rNPC in combination with ILT on cold hypersensitivy were further observed in the second retraining experiment. Animals in the ILT only and the NPC+ILT groups showed significantly increased cold responses during their respective sedentary breaks (*p* < 0.05), while animals in the rNPC group showed very small changes (NS) compared to their initial values prior to breaks. Comparison of sedentary cold response scores showed differences between rNPC+ILT and ILT-only groups, with *p* < 0.01 for the first sedentary period and *p* < 0.05 for the second sedentary period. Cold hypersensitivity scores after the second retraining were also significantly reduced in this recombinant NPC group compared to ILT only (*p* < 0.05).

### 3.3. Intrathecal Injection of Anti-SHG and Bicuculline

Intrathecal injections of bicuculline, SHG antibody, and a combination of SHG antibody and bicuculline were used to assess the contribution of SHG and GABA signaling to the antinociceptive effects of the grafts (Figure 5). The data were normalized to the pre-injection values obtained at the end of 10 week ILT/15 week post-SCI period for each treatment group (dotted line in plots), since these would differ depending on the treatment groups. In addition, the effects of these agents were assessed in the control intact animals and the non-treated SCI animals, and they were also normalized to their own group pre-injection responses.

*Tactile hypersensitivity* (Figure 5a): The intrathecal injection of bicuculline attenuated the effect of NPC transplants in all of the grafted animals with and without ILT (*p* < 0.001 vs. pre-injection), as would be expected for these GABA-producing NPCs. The intrathecal bicuculline also reduced the paw withdrawal latencies in the intact sham animals and exacerbated tactile hypersensitivity in the SCI-only animals compared with their own pre-treatment responses, which was likely due to block of endogenous GABAergic inhibitory processes, as previously reported in [16,29,79,80]. The intrathecal injection of anti-SHG only attenuated the antinociceptive effects of the SHG-producing recombinant grafts (*p* < 0.001 compared with pre-injection values), but had no effects on the non-recombinant graft groups. Comparisons between the corresponding recombinant vs. non-recombinant grafts showed significant differences (*p* < 0.05 between the non-trained cohorts and *p* < 0.001 between the trained cohorts), and also when compared to the pre-injection values (*p* < 0.001). No effects of anti-SHG were observed in either of the control groups. The combined injection of anti-SHG and bicuculline caused an attenuation of the analgesic effects on all of the groups (*p* < 0.001 vs. pre-injection), as would be expected since bicuculline has a broad effect on GABA-A receptors in the spinal cord. When comparing the effects between groups, the animals with recombinant grafts showed a stronger attenuation of analgesia following the combined injection of anti-SHG and bicuculline (*p* < 0.001 compared to groups with non-recombinant grafts), suggesting an additional contribution of SHG from the rNPCs in the anti-allodynic effect. The combined bicuculline and anti-SHG in both of the recombinant groups also produced significantly greater effects on reducing the paw withdrawal thresholds compared with effects on the sham and SCI only controls (*p* < 0.05 and *p* < 0.01 for both vs. rNPC and rNPC+ILT groups, respectively). Interestingly, there was also a significant difference between the trained and non-trained animals in the rNPC group, with a greater apparent reversal of the antinociceptive effects on the rNPC-grafted rats undergoing the ILT (*p* < 0.01).

*Heat hypersensitivity* (Figure 5b): Similar effects of the drugs were observed for heat hypersensitivity, with the attenuation of analgesia being comparable between the groups after the bicuculline injection (*p* < 0.001 compared to pre-injection responses). A stronger decrease in the latencies was observed in the rNPC+ILT group compared to those in the NPC +ILT group (*p* < 0.01). Bicuculline in the NPC group and both the rNPC and rNPC+ILT groups also produced a stronger decrease in their individual pre-treatment latencies compared to the sham or SCI-only groups (*p* < 0.05 vs. NPC, *p* < 0.01 for rNPC and rNPC+ILT). The intrathecal injection of anti-SHG caused the attenuation of antinociceptive effects on only the rNPC animals (*p* < 0.05 and 0.01 for non-trained and trained animals vs. pre-injection values, respectively). Differences in the effectiveness of anti-SHG on reducing the analgesic effects on the NPC+ILT and rNPC+ILT groups following anti-SHG were also observed (*p* < 0.001). Interestingly, comparison within the rNPC group showed that the combined injections had significantly stronger effect on the rNPC+ILT group (*p* < 0.05). With the combined injection of anti-SHG and bicuculline, a significant attenuation of analgesia was observed in all of the groups (*p* < 0.01 for the control groups and *p* < 0.001 for the treatment groups vs. their respective pre-injection values), with there being slightly stronger effects on the animals with recombinant grafts. The reduction was more pronounced in the rNPC+ILT group as compared to those in the NPC+ILT group (*p* < 0.001). The combined injections produced greater effects in terms of reducing the paw withdrawal latencies compared with effects on the sham and SCI only controls (*p* < 0.01 and *p* < 0.001 for both of them vs. rNPC and rNPC+ILT groups, respectively)**.** Stronger effects on reversing antinociception were found in the rNPC+ILT group compared to those on the non-trained rNPC animals (*p* < 0.05).

*Cold hypersensitivity* (Figure 5c): Intrathecal injection of bicuculline attenuated antinociceptive effects on all of the groups (at least *p* < 0.05 vs. pre-injection). Bicuculline in the NPC+ILT, rNPC, and rNPC+ILT groups produced stronger effects in terms of increasing the cold responses from their pre-treatment responses compared with the sham or SCI-only groups (*p* < 0.05, 0.05, and 0.01, respectively). The injection of anti-SHG only attenuated the cold antinociceptive effects on the rNPC groups (*p* < 0.01). The combined injection of bicuculline and anti-SHG caused significantly increased cold responses in all of the animals compared to their pre-injection responses. The reversal of cold antinociceptive effects by the combined bicuculline and anti-SHG was especially robust in the rNPC animals, which resulted in about a 3-fold increase in the responses compared to the pre-injection level (*p* < 0.05 for both NPC groups and sham, and *p* < 0.01 and 0.001 for rNPC and rNPC+ILT vs. pre-injection, respectively). There were also significant differences between the NPC and rNPC groups in both the non-trained and trained animals (*p* < 0.01) and between the controls and rNPC groups in both the non-trained and trained animals (*p* < 0.001).

### 3.4. ELISA Evaluation of Cytokines in Spinal Tissue and CSF

The level of pro- and anti-inflammatory cytokines was evaluated by ELISAs in the lumbar spinal cord (Figure 6) and in the CSF of the animals (Appendix A) at the end of the initial 10 week training period and in a subgroup of animals that were re-trained.

*IL1β:* Spinal cord (Figure 6a): Spinal cord injury produced an upregulation of IL1β in the spinal tissue compared to that of the naïve animals (*p* < 0.001). The cell grafts, ILT, and their combination resulted in a reduction of the IL1β levels compared to the SCI only group, with *p* < 0.01 for the individual ILT, NPC, and rNPC treatments, and *p* < 0.001 for the combined treatments. There were also differences between the treatments, with the most robust differences between the combined treatments of NPC+ILT or rNPC+ILT and their respective grafts without training (*p* < 0.01). These combination treatments nearly reversed the SCI-elevated spinal cord IL1β levels to levels in the non-injured controls (*p* > 0.05 compared with sham). The trained animals in the NPC+ILT and rNPC+ILT groups also showed significantly lower levels of IL1β compared to those of the ILT-only group (*p* < 0.05). The evaluation of IL1β in the spinal cord of retrained and non-retrained cohorts (Figure 6b) showed different outcomes in the animals with and without cell grafts. In the ILT-only group, the spinal IL1β levels were significantly higher in the non-retrained cohort compared to the levels after 10 weeks of ILT or to the retrained cohort (*p* < 0.05). The animals in the NPC+ILT and rNPC+ILT groups showed reduced levels of IL1β compared to the same sets in the ILT-only group (*p* < 0.05 for 10 weeks of ILT and retrained cohort and *p* < 0.001 for non-retrained cohort vs. respective groups in ILT-only group). Unlike the ILT-only group, no differences were observed within or between the animals that underwent the combined treatments.

CSF (Appendix A): The CSF levels of IL1β were reduced in all of the groups compared to SCI-only animals, with strong effects being observed in the ILT-trained animals (*p* < 0.001 vs. SCI only). The ILT training in the grafted animals led to further decrease in IL1β, with *p* < 0.05 for the NPC vs. NPC+ILT and rNPC vs. rNPC+ILT groups. Comparison of the values between the cell-grafted groups showed lower CSF levels of IL1β in the rNPC animals compared to those in the NPC animals (*p* < 0.05). The most robust reduction of CSF IL1β was obtained with the rNPC+ILT group, which further reduced the levels compared to those of the ILT alone (*p* < 0.01). The CSF levels of IL1β in the retrained and non-retrained animals (Appendix A) were comparable within the ILT-only group. Lower levels of IL1β were observed in the grafted animals, with *p* < 0.05 for NPC+ILT vs. ILT only and *p* < 0.01 for rNPC+ILT vs. ILT only for the respective cohorts. Differences were also observed between the retrained and non-retrained cohorts within the respective grafted groups (*p* < 0.05) and also between the retrained cohorts from NPC+ILT and rNPC+ILT groups, with the latter showing the most robust decreases in CSF IL1β (*p* < 0.05).

*TNFα:* Spinal cord (Figure 6c): SCI produced an upregulation of TNFα in the spinal tissue. Either ILT training or cell grafting significantly reduced the TNFα levels compared to those of the SCI-only group (*p* < 0.001). In the animals grafted with rNPC, a significant additional reduction of the spinal TNFα levels was observed between the non-trained and trained subgroups (*p* < 0.05). Retraining the animals helped to preserve the reduced levels of spinal TNFα overall (Figure 6d). Rebound increases in the spinal TNFα following the cessation of the ILT were observed between the retrained and non-retrained cohorts in the ILT-only and NPC+ILT groups (*p* < 0.05). Although there was a trend for the TNFα levels to increase in the non-retrained cohorts of the rNPC+ILT group, this was not significant. Both levels in the non-retrained cohorts from the NPC+ILT and the rNPC+ILT groups were significantly reduced compared to those in the non-retrained ILT-only group (*p* < 0.05).

CSF (Appendix A): The upregulation of TNFα was also observed in the CSF of the SCI-only animals (*p* < 0.001 vs. sham). Cell grafting and the combined grafting with ILT treatment led to a reduction of the CSF TNFα levels compared to those in the SCI-only animals (*p* < 0.05 for NPC and rNPC vs. SCI, *p* < 0.01 for NPC+ILT vs. SCI and *p* < 0.001 for rNPC+ILT vs. SCI). ILT training in combination with cell grafting further reduced the TNFα levels, especially when they were compared to those of the ILT-only treatment (*p* < 0.05 for NPC+ILT and *p* < 0.01 for rNPC+ILT vs. ILT only, respectively). Similar to the CSF IL1β, the most robust reduction of CSF TNFα was in the rNPC+ILT group. Retraining caused reduced levels of CSF TNFα in the animals with the combined treatment in all of the cohorts compared to those of the similarly treated ILT-only sets (Appendix A). The non-retrained cohorts showed rebound increased levels of CSF TNFα in the combined groups compared to the trained cohorts (*p* < 0.05), but the levels of TNFα in these cohorts (non-retrained with combined treatment) were still lower compared to those in the animals who underwent the ILT only (*p* < 0.05).

*IL4:* Spinal cord (Figure 6e): The opposite developments were observed for spinal anti-inflammatory IL4, which was reduced post-SCI (*p* < 0.001 vs. sham) and increased after the treatments, especially in the combined treatment groups with grafts + ILT and rNPC grafting alone (*p* < 0.001 vs. SCI only). In the grafted animals, differences were observed between the NPC and rNPC groups, with there being higher IL4 levels in the rNPC group (*p* < 0.05). ILT training in combination with rNPC grafts showed increased levels of IL4 compared to those in the ILT-only animals (*p* < 0.05). Retraining (Figure 6f) resulted in moderately higher spinal IL4 levels in the retrained cohorts in the rNPC+ILT groups compared to those in the ILT-only group (*p* < 0.05).

CSF (Appendix A): The level of IL4 in the CSF was also reduced after the SCI (*p* < 0.05 vs. sham). Combinations of ILT training with cell grafts elevated the amount of IL4 to levels comparable with those of the sham controls (*p* > 0.05 compared with sham controls), and they were significantly elevated in these combined treatment groups compared with CSF levels following SCI (*p* < 0.05). The CSF levels of IL4 were not affected by retraining (Appendix A), as the values in the retrained and non-retrained cohorts were comparable to their respective initial values after 10 weeks of ILT within each treatment group. There were, however, significant and sustained higher CSF IL4 levels between all of the trained cohorts with cells grafts compared to those of the trained cohorts without grafts (*p* < 0.05).

### 3.5. Immunohistochemical and Biochemical Analysis of Spinal Tissue and CSF

The presence, phenotype, and distribution of grafted cells were evaluated by immunohistochemical staining and FLISA analyses (Figure 7). To identify the grafted cells, some of the non-recombinant cells were transduced with GFP. The recombinant cells were detected by SHG markers. To identify and quantify the phenotype of the grafted cells, double staining with several neuronal and glial markers was used. Figure 7a shows the presence of grafted cells in the spinal dorsal horn. The grafted cells were mostly located within 0.5 mm caudal and rostral from the injection site. On a few occasions grafted cells, identified as GFP or SHG, were observed in the thoracic areas.

The FLISA evaluation of SHG presence in the spinal tissue (Figure 7b) showed high SHG levels in the lumbar segments of the grafted animals, both with and without ILT, compared to those in the thoracic segments (*p* < 0.01). Significantly higher levels of SHG were found in the trained animals compared to those in the non-trained animals (*p* < 0.01). No SHG was detected in the spinal cords of animals receiving non-recombinant NPC grafts. In the CSF, SHG was present in all of the recombinant NPC-grafted animals, with there being higher levels in the trained animals compared with those in the sedentary animals (*p* < 0.05). The amount of SHG was at a background level only in the animals receiving non-recombinant NPCs.

The quantification of the grafts immunocytochemically showed that both the non-recombinant (GFP labeled) and recombinant (SHG labeled) cells survived in larger number in the animals with ILT training compared to those in the non-trained animals, and the majority of these cells developed into GABAergic neurons (Figure 7c, *p* < 0.001 for NPC labeled with GFP; *p* < 0.05 for NPC double labeled with GFP and GABA; Figure 7d, *p* < 0.01 for rNPC labeled with SHG, *p* < 0.05 for rNPC double labeled with SHG and GABA). The total number of GABA-positive cells originating from the grafts were therefore also higher in animals with ILT training compared to those in the non-trained groups. Evaluations of these neuronal markers for GABA and NeuN within the populations of grafted cells showed no differences in the percent of NeuN or GABA cells between the non-trained and trained groups, suggesting that the differentiation of grafted NPCs towards neuronal and GABA phenotypes is not affected by the ILT. With regard to the glial phenotypes, there were minimal differences observed for either GFAP or Iba1 (Figure 7e). Undifferentiated neuronal cells labeled with Doublecortin (Figure 7e) were also essentially unchanged. In general, these phenotypes in the grafted cells were present in insignificant quantities (Figure 7e, GFP/GFAP 8.4% without ILT, 1.6% with ILT; SHG/GFAP 0% for both the trained and nontrained groups; GFP/Iba-1 and SHG/Iba-1 0% for all treatments; GFP/Doublecortin 12.5% without ILT, 7.4% with ILT; SHG/Doublecortin 10.3% without ILT, 8.2% with ILT).

## 4. Discussion

The current study demonstrates the beneficial effects of recombinant SHG-producing GABAergic grafts in combination with locomotor training for the prolonged attenuation of chronic pain after an SCI. For this study, 5 weeks post injury was selected for the initiation of the exercise in order to evaluate the NPC and ILT interventions at time points relevant to the development of the chronic phases of SCI pain. At this time point, the pain symptomology in all three of the neuropathic pain outcome measures in this model had been maximized and stabilized. The exercise rehabilitation strategies in SCI patients vary considerably in many parameters, including time of exercise initiation, intensity, and frequency [81], but intensive locomotor training for at least one month (and up to nearly 2 years) in 196 patients with a clinically incomplete SCI showed significantly improved function when the exercise program was initiated in a wide range from 32 days to over 25 years post-SCI, with there being somewhat less improvement with longer initiation intervals since the SCI [82]. Thus, our ILT start at 5 weeks comports with a feasible and potentially beneficial timeframe for clinical SCIs. Nevertheless, both earlier exercise initiation (as in [41,43,49,54,83]) and later post-ILT initiation with recombinant grafts should be evaluated in future studies.

We have compared the effects of non-recombinant and recombinant grafts in several behavioral tests and evaluated the levels of cytokines in the tissue and CSF. We have also evaluated the long term benefits of such treatments when the training was periodically discontinued, and the animals were re-evaluated for behavioral outcomes after one or two short retraining sessions. Grafts of either naïve or recombinant GABAergic cells attenuated the chronic pain symptoms post-SCI. The recombinant grafts demonstrated generally enhanced capacity in the preservation of the analgesic effects after the discontinuation of training when a more substantial reversal of the beneficial effects was observed in the other treatment groups. Retraining restored the analgesic effects of the treatments to some extent, with there being overall better outcomes in the animals with the recombinant grafts. These data indicate that recombinant GABAergic cell transplants that are capable of both enhancing the inhibitory GABAergic signaling and reducing excitatory NMDA receptor-mediated pathways via SHG release in combination with exercise can potentiate and sustain the attenuation of pain behavior after an SCI.

The beneficial effect of exercise therapy in SCI patients has been reported by numerous clinical and preclinical studies as an effective adjunct therapy to restore mobility and to reduce chronic pain [31,36,39,40,41,42,43,54,83,84]. One of the effects of exercise is an attenuation of the inflammatory responses post-injury. SCI causes an extensive inflammatory reaction not only at the lesion site, but also in more remote areas such as lumbar spinal cord [65]. Hypersensitivity of neurons in this area is related to the development of low-level SCI pain [65,66,85,86], and the involvement of inflammatory mediators in neuronal hyperexcitability has been supported by numerous studies [67,87,88,89]. The attenuation of inflammatory responses by exercise may thus attenuate spinal neuronal hypersensitivity and contribute to pain reduction. Furthermore, the survival of cell grafts targeting this area can be significantly affected by the inflammatory environment [90]. Locomotor training has shown to be effective in attenuating or preventing the development of hypersensitivity in animal models. Previous reports in our lab and others have shown beneficial effects of intensive training and transplantation of GABAergic cells in the restoration of pain-reducing signaling in the spinal cord [31,49]. The beneficial effects of intensive training on intraspinal cell transplants and pain reduction have also been observed in the current study. The analgesic recombinant cells used in combination with locomotor training can potentially maximize the beneficial effects of both of the treatments in reducing SCI pain.

Dysfunctional GABAergic signaling, as one of the key factors underlying neuropathic pain, has been proposed by several studies [9,10,11,12,13,14,15,16,17,18]. The restoration of the inhibitory tone by pharmacological intervention [11,91,92,93] or by cell transplants targeting GABAergic signaling [19,20,21,22,23,24,25,26,27,28,29,30,31] has led to the attenuation of hypersensitivity in animals in various models of chronic pain. Enhanced NMDA signaling also contributes to the development of chronic neuropathic pain [32,94]. Since the clinical use of a potent NMDA antagonists is compromised by their narrow therapeutic window and adverse side effects [95], we have engineered recombinant GABAergic cells which release NMDA antagonist peptide SHG for direct spinal delivery. Previous studies in our lab showed promising antinociceptive effects on neuropathic pain models [20,30]. The results from the current study further support this approach in the treatment of SCI pain, particularly utilizing recombinant SHG-producing GABAergic cells with ILT. These results support the hypothesis that simultaneous modulation of several pain signaling pathways have a highly beneficial value for the management of chronic pain.

To assess the contribution of GABA and/or NMDA antagonist SHG to the beneficial effects of the treatments in reducing SCI neuropathic pain, intrathecal injections of GABA antagonist bicuculline, SHG antibody, or their combination were evaluated for alterations in pain responses. An effect of bicuculline in attenuating antinociception was observed in all of the animals with grafted GABAergic cells, suggesting that GABA released from these cells plays a role in the overall analgesic effect of the graft. Bicuculline also increased pain sensitivity in the intact control animals and in the untreated SCI animals, which is likely due to the role of endogenous dorsal horn GABAergic inhibitory processes in pain modulation [16,29,79,80]. Thus, a component of the beneficial effects of GABAergic cell transplants may be mediated via the restoration of endogenous GABAergic inhibitory processes, as suggested by previous studies indicating contribution from both grafted cells and graft-supported normalization of endogenous GABAergic inhibition [49]. Interestingly, overall, the effects of bicuculline were stronger in the recombinant groups than they were in the controls, suggesting a possible additional role of NMDA antagonism in rescuing the endogenous GABAergic inhibitory function.

The effects of the injection of anti-SHG, on the other hand, were only observed in the animals with recombinant grafts, suggesting the active participation of recombinant SHG in attenuating NMDA signaling and reducing hypersensitivity. The combination of bicuculline with SHG antibody further reduced the beneficial effects of recombinant grafts, as both GABAergic and glutamatergic signaling were targeted. Similar findings have been observed previously using recombinant cell grafts [30]. The current study also suggested that ILT may additionally affect NMDA signaling, as the effects of anti-SHG were more pronounced in the ILT animals than they were in the sedentary animals with recombinant grafts. Spinal attenuation of NMDA subunit 1 expression in rats after exercise has been previously reported [96], and this might underlie the observed increased effects of anti-SHG injection in ILT animals with recombinant grafts in this study. Reduced expression of spinal NMDA receptors in the trained animals could lead to an enhanced contribution of recombinant SHG to the antinociceptive effects compared to those in the non-trained animals. Alternatively, since the release of SHG in the spinal cord of the grafted animals was enhanced in the trained animals compared to that in the non-trained animals, another possibility is that the blocking of the increased SHG levels by the SHG antibody can have a greater impact on the trained vs. untrained animals.

The level of inflammatory mediators was examined in the spinal tissue and CSF of the animals to uncover additional potential mechanisms of the observed effects of our treatments. The biochemical analyses showed reduced levels of proinflammatory cytokines IL1β and TNFα, as well as increased levels of anti-inflammatory IL4, especially in the animals with recombinant grafts. Our previous study showed similar effects of ILT and GABAergic grafts on spinal inflammatory mediators, both in the thoracic (injury level) and lumbar spinal cord segments [49]. A two year study on the same SCI model with maintenance of the ILT protocol showed a sustained reduction of proinflammatory cytokines in correlation with reduced hypersensitivity [97]. These findings comport with the findings of other groups, showing the capability of exercise interventions to reduce chronic inflammation induced by SCIs [44,45,46,47,48]. Together, these data and current data suggest the importance of exercise therapy on the overall outcome related to the attenuation of chronic pain.

In our present study, we have shown that locomotor training in combination with nonrecombinant and recombinant grafts are beneficial for the management of chronic pain. To evaluate the long-term effect of this approach, we discontinued the training for a few weeks, and then we tested for retained antinociceptive effects after the sedentary periods and following the reinstatement of the exercise program. Our results showed that, while the ILT by itself and the ILT in combination with nonrecombinant GABAergic cells are capable of attenuating hypersensitivity in the treated animals, the effects appear to progressively decline after the completion of the training. On the other hand, the recombinant grafts were found to better preserve the benefits of long-term training during an off-training period and to be more readily restored following re-training. Additional interesting results were also found in the small group of animals that underwent secondary re-training. The beneficial effects of graft and training combination appeared to be more potentiated with recombinant cell grafts, which was particularly notable for cold allodynia, when all of the animals showed some reversal of the antinociceptive benefits during the off-training period except for the animals grafted with recombinant cells. The attenuation of mechanical and thermal hypersensitivity was also more effective and stable in the recombinant cell graft groups throughout the repeated off-training and re-training periods. The current study also showed that the grafted animals that underwent re-training had generally lower levels of proinflammatory mediators and higher levels of anti-inflammatory mediators in the lumbar spinal tissue and CSF compared to those of the sedentary animals from the same graft groups. These beneficial effects of ILT retraining in spinal/CSF inflammation were overall more pronounced in the recombinant NPC groups.

Since SCI patients involved in rehabilitation therapy might have limited access or capabilities to participate in daily exercises, an approach that minimizes these requirements and provides stable outcomes could be of significant interest for these patients [98,99,100,101,102,103]. Our study provides a potential strategy to achieve this goal by using recombinant cell grafting in combination with ILT. Other parameters of this approach need to be evaluated to better understand the key variables and potential limitations in development of SCI pain management therapies.

## 5. Conclusions

These results are important regarding the translational value of this experimental therapy, suggesting that intermittent training in combination with targeted cell therapy might be a less demanding yet efficient approach to manage chronic pain in SCI patients.

## Figures and Tables

**Figure 1 bioengineering-10-00084-f001:**
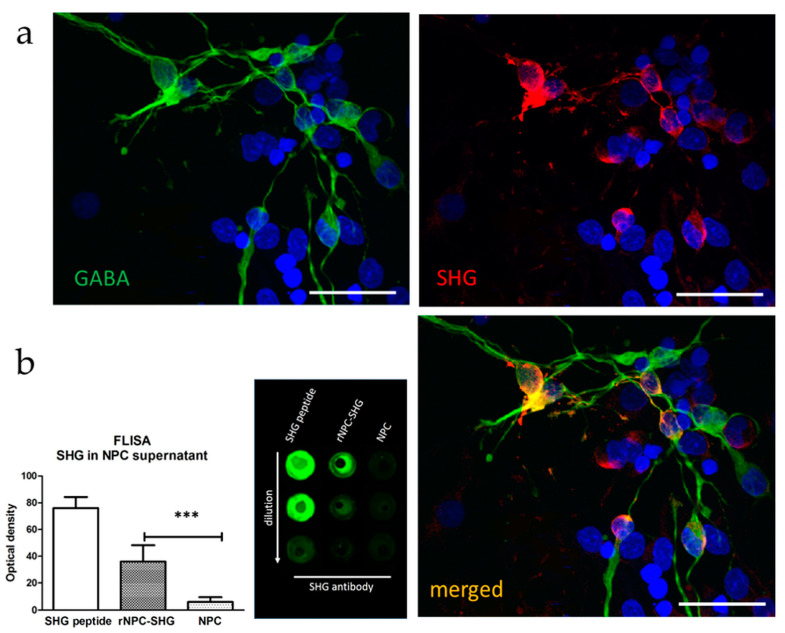
(**a**) Immunocytochemical analysis of transduced E14 rat neuroprogenitors shows colocalization of GABA (green) and SHG (red). Scale bar = 30 µm (**b**) Release of SHG was confirmed by FLISA analysis of cell culture supernatant with SHG peptide as a control. *** *p* < 0.001.

**Figure 2 bioengineering-10-00084-f002:**
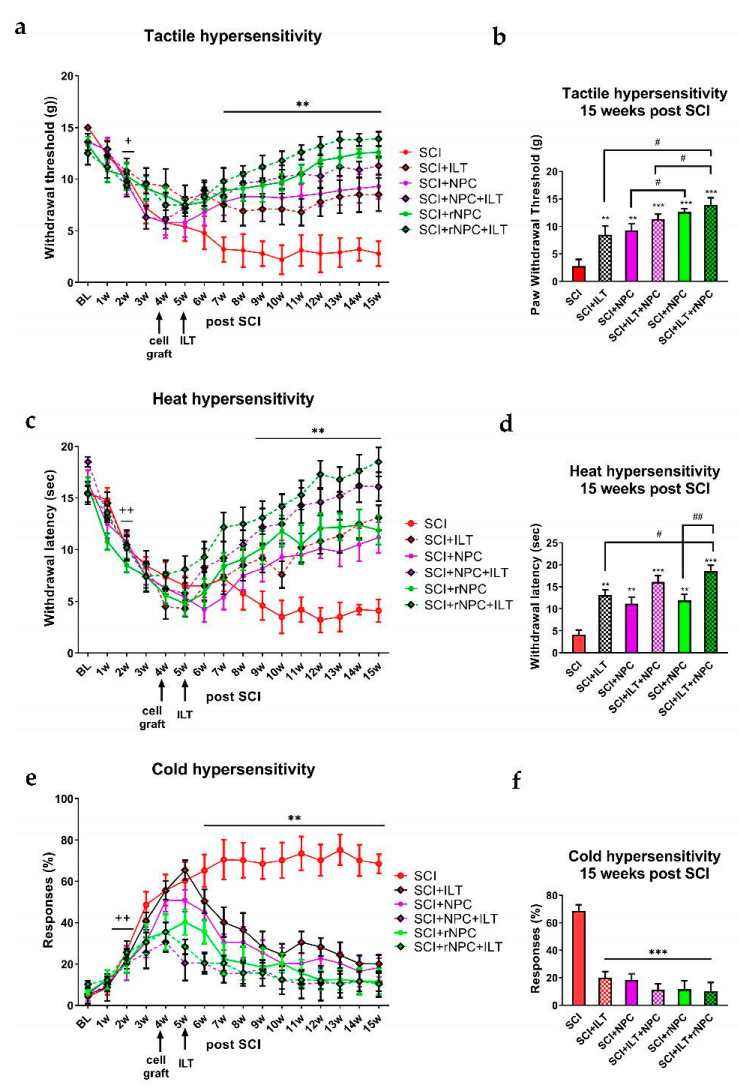
Behavioral evaluation of tactile (**a**,**b**), heat (**c**,**d**), and cold (**e**,**f**) hypersensitivity in SCI animals who underwent different treatments. (**a**,**c**,**e**) For better clarity of time-course plots, statistical marks are indicated over time points where all animals showed significantly different values compared to the baseline or SCI only group. + *p* < 0.05, ++ *p* < 0.01 vs. baseline, ** *p* < 0.05 vs. SCI only. (**b**,**d**,**f**) Values after 10 weeks of ILT/15 weeks of SCI were compared between all groups. ** *p* < 0.01, *** *p* < 0.001 vs. SCI, # *p* < 0.05, ## *p* < 0.01between indicated groups. N = 10 (SCI, NPC, rNPC); N = 16 (ILT, ILT+NPC, ILT+rNPC).

**Figure 3 bioengineering-10-00084-f003:**
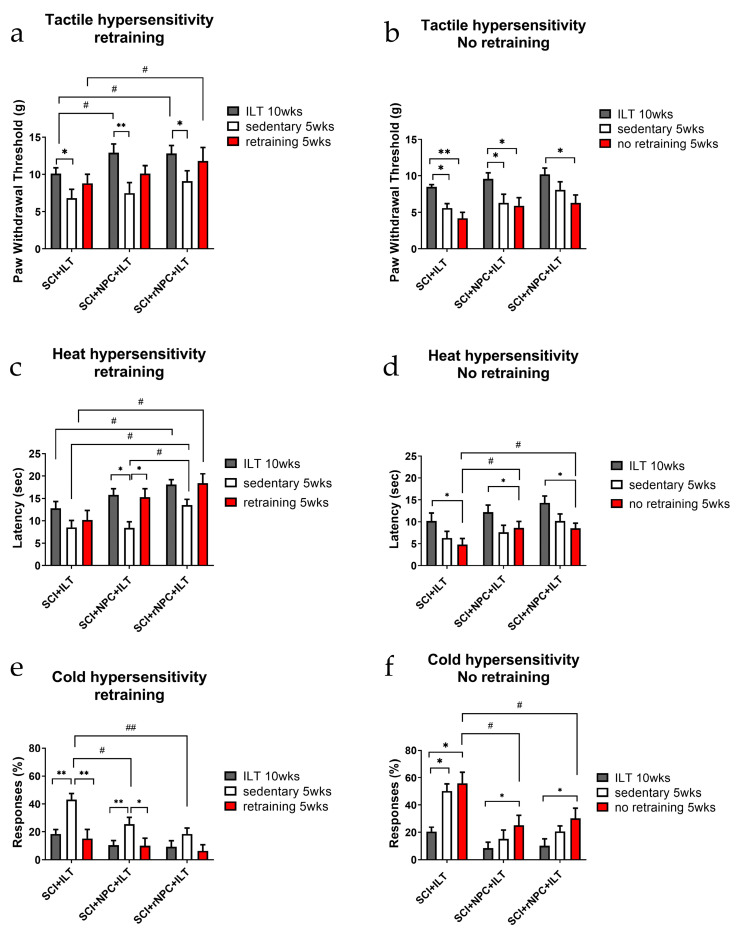
Evaluation of tactile (**a**,**b**), heat (**c**,**d**), and cold (**e**,**f**) hypersensitivity in retrained (left column) and sedentary (right column) animals in SCI+ILT, SCI+NPC+ILT, and SCI+rNPC+ILT groups. * *p* < 0.05, ** *p* < 0.01 between training/sedentary/retraining periods within each treatment group. # *p* < 0.05, ## *p* < 0.01 between respective time points of different groups. N = 6/group in retrained cohort; N = 4/group in non-retrained cohort.

**Figure 4 bioengineering-10-00084-f004:**
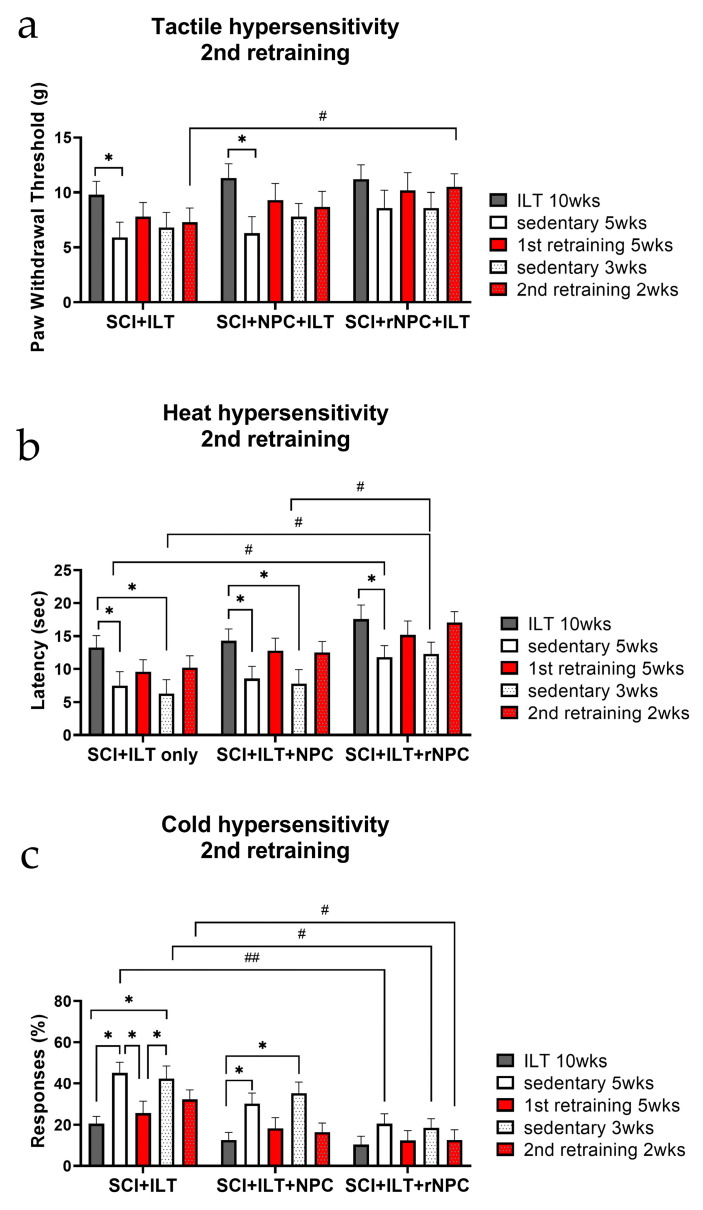
Evaluation of tactile (**a**), heat (**b**), and cold (**c**) hypersensitivity in animals that underwent a second retraining in SCI+ILT, SCI+NPC+ILT, and SCI+rNPC+ILT groups. * *p* < 0.05 between different training periods within each treatment group. # *p* < 0.05, ## *p* < 0.01 between respective training periods of different groups. N = 4/group.

**Figure 5 bioengineering-10-00084-f005:**
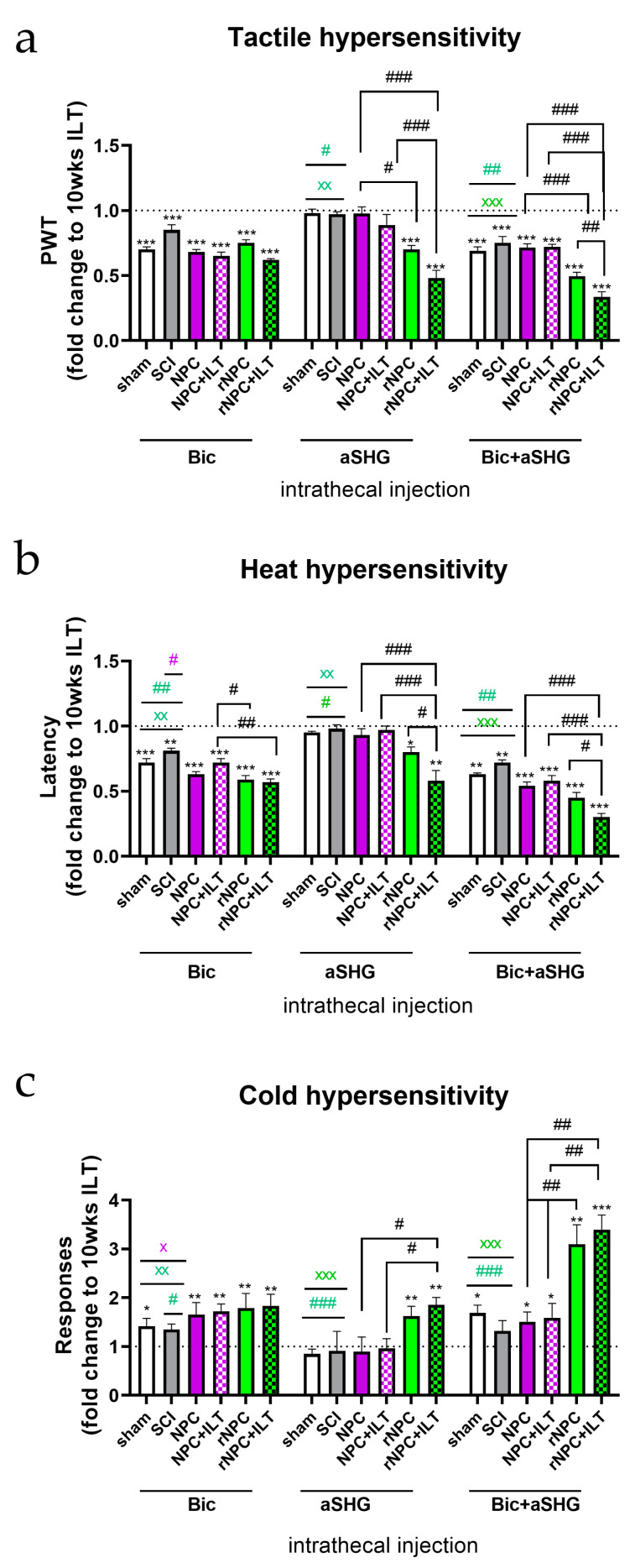
The effect of intrathecal injection of bicuculline (Bic), anti-SHG (aSHG), and their combination on tactile (**a**), heat (**b**), and cold (**c**) hypersensitivity in animals with naïve and recombinant cell graft with and without ILT. Data were normalized to pre-injection values (10 weeks of ILT, indicated by dotted line), and they are expressed as fold change. * *p* < 0.05, ** *p* < 0.01, *** *p* < 0.001 vs. 10 wks ILT; color coded # *p* < 0.05, ## *p* < 0.01, ### *p* < 0.001 between indicated groups (black), vs. rNPC (green), vs. NPC (purple); color coded ^x^
*p* < 0.05, ^xx^
*p* < 0.01, ^xxx^
*p* < 0.001 vs. rNPC+ILT (green), vs. NPC+ILT (purple). N = 6–7/group.

**Figure 6 bioengineering-10-00084-f006:**
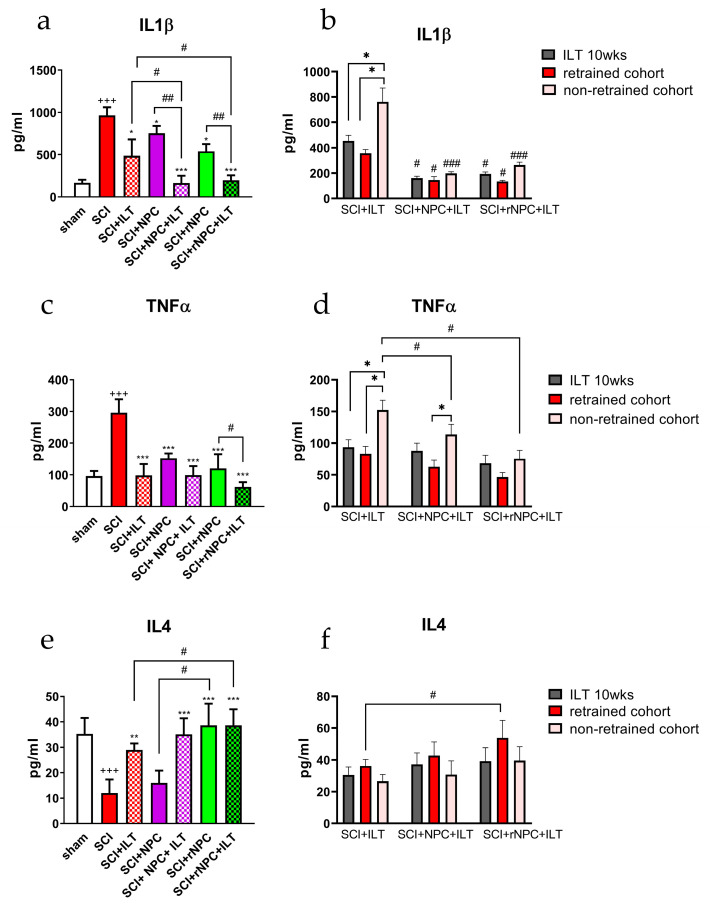
ELISA analysis of IL1β (**a**,**b**), TNFα (**c**,**d**), and IL4 (**e**,**f**) in the spinal cord of animals with different treatments after 10 weeks of ILT/15 weeks post-SCI (left column) and in retrained and non-retrained cohorts (right column). (**a**,**c**,**e**) +++ *p* < 0.001 vs. sham, * *p* < 0.05, ** *p* < 0.01, *** *p* < 0.001 vs. SCI, # *p* < 0.05, ## *p* < 0.01, ### *p* < 0.001 between indicated groups. (**b**,**d**,**f**) * *p* < 0.05 during training/retraining/sedentary periods within each treatment group. # *p* < 0.05, ### *p* < 0.001 for SCI+NPC+ILT and SCI+rNPC+ILT vs. respective training periods in SCI+ILT group and between indicated groups. N = 3–4/group.

**Figure 7 bioengineering-10-00084-f007:**
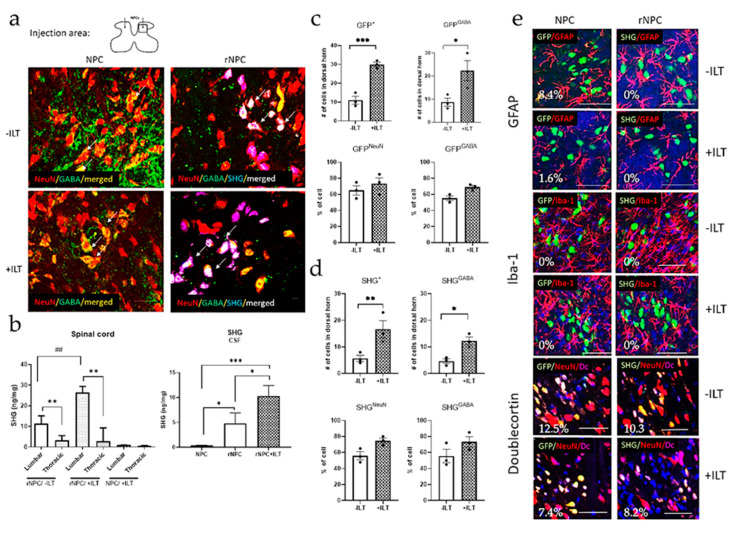
Immunohistochemical analysis of spinal tissue of grafted animals. (**a**) NPCs (GABAergic, left column) and rNPCs (GABA/SHG, right column) were injected as depicted. Tissue from sedentary (−ILT, top row) and trained (+ILT, bottom row) animals were compared for the presence of grafted cells. Arrows indicate double and triple labelled cells with NeuN (red), GABA (green) and SHG (blue) antibodies. Scale bar = 30 µm. (**b**) FLISA analysis of SHG in spinal tissue and CSF in trained and non-trained animals with cell grafts. N = 3–4/group. (**c**,**d**) Quantification of NPC grafted cells labeled by GFP (**c**) or SHG (**d**) and their colocalization with NeuN or GABA markers in trained and non-trained animals. Each marker was analyzed on at least 10 sections. (**e**) Colocalization of grafted cells (labeled by GFP or SHG, both in green) with glial phenotypes (GFAP or Iba1, both in red) or immature proneuronal cells (doublecortin in magenta, NeuN in red, GFP/SHG in green). The percentage of colocalization with GFP or SHG positive cells are indicated. Scale bar = 50 µm. * *p* < 0.05, ** *p* < 0.01, *** *p* < 0.001 between indicated groups; ## *p* < 0.01 between indicated groups.

## Data Availability

All data are available upon request from corresponding author.

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
