# Peer review of "Attenuation of SCI-Induced Hypersensitivity by Intensive Locomotor Training and Recombinant GABAergic Cells"

_bioengineering, 2023, doi:10.3390/bioengineering10010084_

Round 1

Reviewer 1 Report

We read with great interest the article by Jergova et al titled  “attenuation of SCI-induced hypersensitivity by intensive locomotor training and recombinant GABAergic cells” where the authors evaluated the dual intervention of using recombinant GABAergic cells releasing NMDA blocker serine-histogranin 9 Coupled with intensive locomotor training (ILT) to ameliorate chronic pain in SCI.

The study has exciting findings especially related to the fact that repeated training is needed for comprehensive rehabilitation. This work builds on previous work by the same group; however, there are some comments that need to be addressed:

Comments:

The author did not justify the n used for their experiments

They need to provide a full power analysis section used.

The work involved 78 rats, the authors should indicate the exact cohorts in their study with their numbers involved. Also, they should indicate how many animals were used for each endpoint presented

The author selected 5 weeks for their exercise to start, can they put this time frame of clinical SCI and should provide a description for this timeline used in starting exercise at chronic time points post injury and how it correlates to rehabilitation in humans post TBI or SCI?

For the NPC selection, the authors should perform immunostaining with glutamate to show that their NPCs do not contain any glutaminergic cells.

Major comments:

 Several experiments lacked proposer controls such as in figures: 5 and 6 where control animals with saline and no SCI group and a group with SCI should be included. That would give a better assessment of the treatment.

For the IF, assessment of GFAP and Iba1 should be assessed among the different groups, this is different from what the authors are showing in Figure 7;

This should be also included in the animals treated with anti-SHG alone and/or bicuculline-injected rats

It would be important if these interventions affected inflammatory cells in the injury site or not

Also, the authors should evaluate if there was any glial scarring post injury 

Author Response

Thank you for your insightful comments and suggestions. We have addressed issues raised in your comments as follows:

We read with great interest the article by Jergova et al titled  “attenuation of SCI-induced hypersensitivity by intensive locomotor training and recombinant GABAergic cells” where the authors evaluated the dual intervention of using recombinant GABAergic cells releasing NMDA blocker serine-histogranin 9 Coupled with intensive locomotor training (ILT) to ameliorate chronic pain in SCI.The study has exciting findings especially related to the fact that repeated training is needed for comprehensive rehabilitation. This work builds on previous work by the same group; however, there are some comments that need to be addressed:

Comments:

The author did not justify the n used for their experiments. They need to provide a full power analysis section used.

The work involved 78 rats, the authors should indicate the exact cohorts in their study with their numbers involved. Also, they should indicate how many animals were used for each endpoint presented

Response:

The power analysis and justification for number of animals in the study has been added into the Method section (lines 81-87). The exact number of animals/samples per group for each endpoint is included in the corresponding figure legends.

The author selected 5 weeks for their exercise to start, can they put this time frame of clinical SCI and should provide a description for this timeline used in starting exercise at chronic time points post injury and how it correlates to rehabilitation in humans post TBI or SCI?

Response:

We have added clarification about the selected time points for SCI and training in the Discussion (lines 693-705). Other time points are suggested as future directions.

For the NPC selection, the authors should perform immunostaining with glutamate to show that their NPCs do not contain any glutaminergic cells.

Response:

Glutamate staining of GABAergic NPCs has been added as new Supplementary figure 1. Few, if any glutamatergic NPCs were found.  This may be due to our standard differentiation process using withdrawal of FGF2 which enhances GABAergic NPC differentiation.

Major comments:

Several experiments lacked proposer controls such as in figures: 5 and 6 where control animals with saline and no SCI group and a group with SCI should be included. That would give a better assessment of the treatment.

Response:

The data for the control groups (sham without SCI and SCI only) have been included as suggested in the revised Figure 5, as well as included in the results description (lines 487-488, 491-495, 500, 507-510, 518-520, 530-532, 536-539, 547-548) and discussion (lines 756-764) sections.  In Figure 6, the levels of inflammatory mediators were already included for all groups (including sham without SCI and SCI only) in the initial 10 week study.  Since only animals undergoing re-training/sedentary were used in later studies, only those animals included and available from that part of the study could be evaluated for inflammatory mediators.    

For the IF, assessment of GFAP and Iba1 should be assessed among the different groups, this is different from what the authors are showing in Figure 7.

Response:

Figure 7 has been updated to include photomicrographs and quantification of GFAP and Iba in these different groups according to this suggestion (lines 677-689)

This should be also included in the animals treated with anti-SHG alone and/or bicuculline-injected rats.

Response:

Treatment with anti-SHG and bicuculline is an acute evaluation and both agents are short-lasting (effects dissipate within a few hours and return to pre-injection pain responses). In addition, following 48 washouts, animals received the different drug treatments, as described in the methods.  Thus, we would not expect to see any differences or long-term changes in cellular phenotypes and this was not evaluated.  

It would be important if these interventions affected inflammatory cells in the injury site or not.  Also, the authors should evaluate if there was any glial scarring post injury.

Response:

We have not evaluated the effect of the treatment on the level/expression of inflammatory mediators or glial scarring at the injury site in the study, as this was beyond the current scope.  We have previously reported ELISA evaluation of cytokine levels from the injury site in a previous study with non-recombinant NPCs and ILT (Dugan et al., 2020).  We agree that injury site data would provide further insights on effects of these interventions in SCI to incorporate in future evaluations.

Reviewer 2 Report

The present manuscript was interesting for readers. Some errors should be revised.

Line 73, the reference (Kilkenny et al., 2010) should be presented as the number.

Figures were too small to read the graph labels in printed version. Only when PDF file for review was zoomed upto 200-400% in PC, the word could be read.

In Figure 6, some ** and *** in graphs had errors.

Line 616-621 and Figure 7c, the images shown in figure 7c were DAPI/NeuN, DAPI/GFP, DAPI/Iba-1, and merged. The images with GFAP and doublecortin immunostainings were lost. And, which were with ILT and without ILT?

There were 93 references in the present manuscript and 26 among them might be authors' own. The number might be too many. Some references in methods could be reduced.

Author Response

Thank you for your comments and recommendations, we have addressed and corrected issues raised in your review as follows:

The present manuscript was interesting for readers. Some errors should be revised.

Line 73, the reference (Kilkenny et al., 2010) should be presented as the number.

Response:

This has been fixed in the revision. 

Figures were too small to read the graph labels in printed version. Only when PDF file for review was zoomed upto 200-400% in PC, the word could be read.

Response:

Thank you for pointing this out. The figures were corrupted in the previous version and we have now increased the resolution. Original high resolution figures are also uploaded to the submission system  in zip folder.

In Figure 6, some ** and *** in graphs had errors.

Response:

This has been fixed in the revised Figure 6.

Line 616-621 and Figure 7c, the images shown in figure 7c were DAPI/NeuN, DAPI/GFP, DAPI/Iba-1, and merged. The images with GFAP and doublecortin immunostainings were lost. And, which were with ILT and without ILT?

Response:

The results section and the corresponding Fig. 7 plates have been revised and additional images have been included to show the GFAP and doublecortin and the with/without ILT (lines 677-689).

There were 93 references in the present manuscript and 26 among them might be authors' own. The number might be too many. Some references in methods could be reduced.

Response:

We have reduced the number of references particularly from our own group.  However, some additional references were needed in response to reviewer comments.

Reviewer 3 Report

The topic of this article is interesting, the authors presenting the methodology and the results of the use of recombinant GABAergic cells releasing NMDA blocker serine-histogranin, and the intermittent training on the hyperalgesia induced after spinal cord injury in rats.

After reading the manuscript, the following doubts and suggestions have arisen.

The abstract should be redone, to state that the researches are conducted on rats.

The introduction section should be more complete, providing supplementary background in the field.

The results obtained should be compared with those achieved by other researchers and discussions should be significantly detailed.

In discussion section, the authors need to develop argumentation in depth based on the current understanding and the findings of the results obtained, presenting the potential, the weakness and limitation, and future research direction, among others. Authors should try to explain the theoretical implication as well as the translational application of their research.

Some other aspects were found in this manuscript:

- all abbreviations should be expanded in the first appearance. The explanation of the abbreviation should be used only once in the text and should not be repeated, in order to decongest the text and facilitate the understanding of the information transmitted

- many of the abbreviations are not explained, which creates a lot of confusion and makes the text difficult to read and understand: GABA,NMDA, BBB, DAPI, FLISA, ELISA, IL-1β, TNF-α, CSF, CO2, BCA, GFP, CCI and others;

- different fonts were used in the text and in the figures;

- the authors should improve the quality of some figures;

- missing information (city, country) about the companies producing some devices used in the research (lines: 84, 90, 115, 133, 142, 161, 174, 185, 234 and others);

- the author Kilkenny et al., 2010 mentioned on line 73 is missing from the references

- at the references the authors should provide the DOI of the articles

- a schematic representation of the study would be appreciated;

- spelling check of the text is mandatory;

- English including grammar, style and syntax, should be improved through the professional help from English Editing Company for Scientific Writings.

Author Response

Thank you for your kind words and recommendations, we have addressed, discussed, and corrected issues raised in your review as follows:

The topic of this article is interesting, the authors presenting the methodology and the results of the use of recombinant GABAergic cells releasing NMDA blocker serine-histogranin, and the intermittent training on the hyperalgesia induced after spinal cord injury in rats.

After reading the manuscript, the following doubts and suggestions have arisen.

The abstract should be redone, to state that the researches are conducted on rats.

Response:

The abstract has been re-stated (with the respect to word limit) to clearly indicate that study was performed on rats.

The introduction section should be more complete, providing supplementary background in the field.

Response:

More background information has been added to the Introduction (lines 53-57, 61-65).

The results obtained should be compared with those achieved by other researchers and discussions should be significantly detailed.

In discussion section, the authors need to develop argumentation in depth based on the current understanding and the findings of the results obtained, presenting the potential, the weakness and limitation, and future research direction, among others. Authors should try to explain the theoretical implication as well as the translational application of their research.

Response:

The discussion has been updated to include more in depth understanding of our results in the context of other findings, and also to discuss limitations, translational application and future directions (lines 693-705, 756-764, 785-792, 807-820). 

Some other aspects were found in this manuscript:

- all abbreviations should be expanded in the first appearance. The explanation of the abbreviation should be used only once in the text and should not be repeated, in order to decongest the text and facilitate the understanding of the information transmitted

- many of the abbreviations are not explained, which creates a lot of confusion and makes the text difficult to read and understand: GABA,NMDA, BBB, DAPI, FLISA, ELISA, IL-1β, TNF-α, CSF, CO2, BCA, GFP, CCI and others;

Response:

The abbreviations have all been explained as suggested. 

- different fonts were used in the text and in the figures;

Response:

The fonts have been corrected according to the template.

- the authors should improve the quality of some figures;

Response:

The resolution of the images and font size have been corrected for better clarity. Original high resolution figures are also uploaded to the submission system  in zip folder.

- missing information (city, country) about the companies producing some devices used in the research (lines: 84, 90, 115, 133, 142, 161, 174, 185, 234 and others);

Response:

This information has been added as suggested.

- the author Kilkenny et al., 2010 mentioned on line 73 is missing from the references

Response:

This has been corrected.

- at the references the authors should provide the DOI of the articles

Response:

We have added DOI where available. For references, we are using the format suggested by the editorial office, and further corrections will also be made according to their requirements.

- a schematic representation of the study would be appreciated;

Response:

A schematic representation and timeline of the study has been added as a Supplementary figure 1 as suggested.

- spelling check of the text is mandatory;

- English including grammar, style and syntax, should be improved through the professional help from English Editing Company for Scientific Writings.

Response:

Spelling, grammar, style, and syntax have been checked and improved.

Round 2

Reviewer 1 Report

Accept in present form

Reviewer 3 Report

The authors mostly responded to the comments and suggestions and the manuscript was revised accordingly. I consider it could be accepted for publication in this journal, but I propose to have the manuscript checked by a native English speaking person.